# Trypsin from Pyloric Caeca of Asian Seabass: Purification, Characterization, and Its Use in the Hydrolysis of Acid-Soluble Collagen

**DOI:** 10.3390/foods12152937

**Published:** 2023-08-02

**Authors:** Umesh Patil, Khurshid Ahmed Baloch, Shivraj Hariram Nile, Jun Tae Kim, Soottawat Benjakul

**Affiliations:** 1International Center of Excellence in Seafood Science and Innovation, Faculty of Agro-Industry, Prince of Songkla University, Hat Yai 90110, Songkhla, Thailand; umesh.p@psu.ac.th (U.P.); khurshidbaloch555@gmail.com (K.A.B.); 2Division of Food and Nutritional Biotechnology, National Agri-Food Biotechnology Institute (NABI), Sector 81, Sahibzada Ajit Singh Nagar 140306, Punjab, India; shivraj.nile@nabi.res.in; 3Department of Food and Nutrition, Kyung Hee University, Seoul 02447, Republic of Korea; jtkim92@khu.ac.kr

**Keywords:** purification, trypsin, biochemical characterization, hydrolysis, collagen

## Abstract

The study aimed to purify trypsin from the pyloric caeca of Asian seabass (*Lates calcarifer*), and investigate its proteolytic capability toward acid-soluble collagen (ASC) in comparison with commercial porcine trypsin (CPT). Trypsin was purified from pyloric caeca, a leftover from the evisceration process, via ammonium sulphate (40–60% saturation) precipitation, and a soybean trypsin inhibitor (SBTI)–Sepharose 4B column. A 18.5-fold purification and a yield of 15.2% were obtained. SDS-PAGE analysis confirmed a single band of trypsin with a molecular weight of 23.5 kDa. Purified trypsin also showed the single band in native-PAGE. The optimal pH and temperature of trypsin for BAPNA (the specific substrate for amidase) hydrolysis were 8.5 and 60 °C, respectively. The trypsin was stable within the pH range of 7.0–9.5 and temperature range of 25–55 °C. Protease inhibition study confirmed that the purified enzyme was trypsin. The purified trypsin had a Michaelis–Menten constant (K_m_) and catalytic constant (k_cat_) of 0.078 mM and 5.4 s^−1^, respectively, when BAPNA was used. For the hydrolysis of TAME (the specific substrate for esterase), the K_m_ and K_cat_ were 0.09 mM and 4.8 s^−1^, respectively. Partially purified seabass trypsin (PPST) had a slightly lower hydrolysis capacity toward ASC than CPT, as evidenced by the lower degree of hydrolysis and protein degradation when the former was used. Both the α-chain and β-chain became more degraded as the hydrolysis time increased. Based on MALDI-TOP, peptides with MW of 2992-2970 Da were dominant in the hydrolysates. Therefore, seabass trypsin could be used in the production of hydrolyzed collagen. It could have economic importance to the market, by replacing some commercial proteases, which have religious constraints.

## 1. Introduction

Proteases have been extensively used in the food industries for the hydrolysis of food proteins to peptides with enhanced bioactivities and functional properties [1]. The high prices and religious restrictions of porcine and bovine proteases are still the main concerns regarding their applications [2]. Nevertheless, the consumption of marine organisms has no religious limitations, and several products, such as enzymes or bioactive peptides, etc., have attracted attention from the food, biomedical, cosmetic, nutraceutical, and pharmaceutical industries [3]. Polyunsaturated fatty acids have been extracted from tuna eyeball oil [4], and minerals have been produced from shrimp waste [5]. In addition, other active compounds with different nutritive values and biological activities can be extracted from various marine sources [6,7]. Thus, the rising demand for value-added biochemicals and pharmaceuticals from fish leftover has brought about an increasing interest among researchers and entrepreneurs [8,9].

In Thailand, Asian seabass (*Lates calcarifer*) is an important species, for both local consumption and export. During fish processing or dressing, a large amount of byproduct (~0.4 million tons of by-product/year), including viscera, is produced, and turns to wastage, which can cause pollution if its treatment or management is not implemented properly [10]. Viscera are inedible, and are generally removed during evisceration. Trypsin and pepsin are the two major proteolytic enzymes found in fish viscera [2]. Trypsin (alkaline digestive proteases) has high activity in the alkaline pH range (8.0–10.0), whereas pepsin (acidic proteases) is most active between pH 2.0 and 4.0 [11]. Trypsin is a very important enzyme, with several commercial uses in food companies. Trypsin has been isolated, purified, and characterized from various species of fish, including the mrigal carp (*Cirrhinus mrigala*) [12], common kilka (*Clupeonella cultriventris caspia*) [13], pirarucu (*Arapaima gigas*) [14], small red scorpion fish (*Scorpaena notata*) [15], albacore tuna (*Thunnus alalunga*) [16], catfish (*Luphiosilurus alexandri*) [17], gulf corvina (*Cynoscion parvipinnis*) [18], Monterey sardine (*Sardinops sagax caerulea*) [19], common dolphinfish (*Coryphaena hippurus*) [20], skipjack tuna (*Katsuwonus pelamis*) [21], zebra blenny (*Salaria basilisca*) [22], sardinelle (*Sardinella aurita*) [23], Atlantic bonito (*Sarda sarda*) [24], tambaqui (*Colossoma macropomum*) [25], yellowfin tuna (*Thunnus albacores*) [26], beluga (*Huso huso)*, and sevruga (*Acipenser stellatus*) [27]. Trypsin was used to hydrolyze gelatin from the skin of giant catfish, in which antioxidative peptides were generated [28]. In addition, trypsin from skipjack tuna spleen was stabilized using Na-alginate and chitosan (low molecular weight) beads [29]. It was used as a digestion aid, along with tuna pepsin, in a simulated gastrointestinal tract [2].

Traditionally, the pyloric caeca is ground together with other organs into fish flour, which is subsequently utilized as feed for animals with a low market value [30]. To better exploit fish viscera, proteases have been extracted, and used as a potential alternative, due to their abundance. The pyloric caeca from Asian seabass can be a promising source of enzymes, particularly trypsin, which can be applied in protein hydrolysate preparation or for hydrolysis purposes in the food industry, to lower the enzyme cost, and avoid religious limitations [2]. However, the molecular property of trypsin from the pyloric caeca of Asian seabass has not been documented. Moreover, trypsin isolated from the leftover of the fish-processing industry has no religious restrictions. This aspect or advantage broadens the potential applications of fish trypsin in various industries, including food and medicine, etc. The proteolytic ability of trypsin from the pyloric caeca of Asian seabass toward several proteins, such as fish muscle, milk, and legume proteins, has been reported [2]. Nevertheless, no hydrolytic ability of trypsin from Asian seabass pyloric caeca toward acid-soluble collagen exists. The resulting hydrolyzed collagen has been known for skin nourishment, bone strengthening, and other bioactivities [31,32]. Therefore, the aims of this study were to extract and characterize trypsin from the pyloric caeca of Asian seabass, and to compare its proteolytic ability toward acid-soluble collagen with commercial porcine trypsin.

## 2. Materials and Methods

### 2.1. Chemicals and Materials

All the chemicals used were of analytical grade, and procured from Sigma (St. Louis, MO, USA). The viscera of Asian seabass were removed from deceased fish sacrificed by a vendor in Koyo Island, Songkhla. The freshly removed viscera were placed in a polyethylene bag, and transported in ice to the laboratory within 30–40 min.

### 2.2. Preparation of Pyloric Caeca Crude Extract

The pyloric caeca were separated from the viscera, and cut into small pieces. In liquid nitrogen, pieces of pyloric caeca were blended, until a uniform powder was obtained. For defatting, the powder was added with chilled acetone at 1:3 (*w*/*v*). The mixture was then homogenized for 5 min at 15,000 rpm and 4 °C, with an IKA Labortechnik homogenizer (Selangor, Malaysia). The overhead stirrer was used to stir the mixture at 4 °C for 40 min at 300 rpm. Then, the mixture was vacuum-filtered using Whatman paper No. 4, and the filtrate was air-dried for 24 h. The dried Asian seabass pyloric caeca (ASPC) powder was immediately used for crude trypsin extraction. The method of Khantaphant and Benjakul [33] was adopted to prepare the crude trypsin from the ASPC. Briefly, the ASPC powder was added to ten volumes of extraction buffer, 50 mM Tris-HCl buffer (pH 8.0) containing 10 mM CaCl_2_. The mixture was homogenized for 2 min, and then stirred at 300 rpm and 4 °C for 1 h. The homogenate was centrifuged at 8000× *g* and 4 °C for 30 min. The supernatant, named as ‘ASPC extract’, was further used for further purification.

### 2.3. Purification of Trypsin

#### 2.3.1. Preparation of Ammonium Sulphate Fraction

The ASPC extract was subjected to ammonium sulphate precipitation (40–60% saturation). Ammonium sulphate power was gradually added into the ASPC extract, under stirring conditions at 4 °C for 30 min. Afterward, the mixture was centrifuged at 7000× *g* and 4 °C for 30 min. The pellets were collected, and dissolved in the minimal volume of 50 mM Tris-HCl buffer, pH 8.0. Subsequently, the solution was further dialyzed overnight, against ten volumes of extraction buffer, at 4 °C. The dialysis buffer was replaced three times, and then the dialysate was collected, and referred to as ‘partially purified seabass trypsin’ or ‘PPST’.

#### 2.3.2. Purification of Trypsin

An affinity column was packed with soybean trypsin inhibitor (SBTI)–Sepharose 4B [34]. The Sepharose 4B was firstly washed and activated with 1 mM HCl at 1:200 (*w*/*v*). The SBTI and Sepharose (1:0.027, *w*/*w*) were coupled at room temperature. The remaining active groups in the coupling medium were blocked and deactivated with 0.1 M Tris-HCl buffer (pH 8.0). PPST was added to SBTI–Sepharose 4B column (1 × 10 cm). The column was pre-equilibrated with 50 mM Tris-HCl buffer (pH 8.0) containing 10 mM CaCl_2_ and 0.5 M NaCl. Thereafter, the column was washed with the same buffer, without CaCl_2_, at a flow rate of 0.5 mL/min. The column was washed several times, until the absorbance at 280 nm was less than 0.005. Then, the sample was eluted, using 5 mM HCl at a flow rate of 1.0 mL/min. Different fractions (2 mL) were collected, and rapidly mixed with 0.5 mL of 100 mM Tris-HCl buffer, pH 8.5, containing 10 mM CaCl_2_. All the fractions were analyzed for trypsin activity, using BAPNA as the substrate. In addition, the absorbance of 280 nm (OD_280_) was read for all the fractions. The curves of fraction number versus activity, and fraction number versus OD_280_ were plotted. The fractions with a high activity were pooled, and further dialyzed against ten volumes of 10 mM Tris-HCl buffer (pH 8.0) containing 1 mM CaCl_2_ at 4 °C. The dialysis was conducted overnight, and the dialysis buffer was changed three times during dialysis. The fraction was referred to as ‘purified trypsin’, or PT.

### 2.4. Trypsin Activity Assay

The trypsin activity was determined as per the method of Patil and Benjakul [35]. The BAPNA substrate (2 mg/mL) was prepared in DMSO. The reaction mixture was prepared by mixing 200 µL of sample with 1 mL of reaction buffer (50 mM Tris-HCl buffer pH 8.0 with 10 mM CaCl_2_). BAPNA (200 µL) was added to the reaction mixture, to initiate the reaction. The hydrolysis of BAPNA took place for 20 min at 60 °C. Afterward, 200 µL of 30% acetic acid (*v*/*v*) was added, to stop the reaction. The release of free *ρ*-nitroaniline was measured from the absorbance at 410 nm. The blank was conducted in the same manner, but the trypsin was deactivated with 30% acetic acid, before the addition of the substrate. One unit was defined as trypsin which released *ρ*-nitroaniline at 1 nmole/min. The equation for trypsin activity calculation was given below:$$Trypsin\;activity=\frac{(A-A_{0})\times mixture\;volume\;(mL)\times 1000}{8800\times reaction\;time\;\left(min\right)\times 0.2}$$
where *A* represents the absorbance of experimental groups, and *A*_0_ represents the absorbance of blank, and 8800 (cm^−1^ M^−1^) is the extinction coefficient of *ρ*-nitroaniline.

### 2.5. Characterization of Trypsin

#### 2.5.1. Sodium Dodecyl Sulphate–Polyacrylamide Gel Electrophoresis (SDS-PAGE) and Substrate–Gel Electrophoresis

The molecular weights (MWs) of the PPST and PT were determined using SDS-PAGE [36]. The samples were mixed with the sample buffer containing 125 mM Tris-HCl, SDS (4%; *w*/*v*), and glycerol (20%; *v*/*v*), at a ratio of 2:1 (*v*/*v*). A protein (12 µg) from the PPST and PT was applied to the gel (12% separating gel and 4% stacking gel). Mini-Protein II electrophoresis (Bio-Rad, Hercules, CA, USA), at a constant current of 15 mA, was used for the protein separation. Native-PAGE was performed using 12% separating gel, in a similar manner, except that the sample was not heated, and all solutions contained no SDS or any reducing agent. The gels were stained with 0.05% Coomassie Blue G-250, and 15% methanol solution containing 5% acetic acid, followed by destaining.

Substrate gel electrophoresis was carried out, as per the method of Senphan et al. [37]. The gels, after electrophoresis, were immersed in 100 mL of 50 mM Tris-HCl buffer (pH 7.5) containing 2% casein, at 4 °C for 1 h, under mild shaking conditions. Afterward, the gels were placed in a substrate mixture (50 mM Tris-HCl buffer, pH 8.5 with 2% casein), and incubated at 60 °C for 15 min, with continuous gentle shaking. The gels were then stained and destained, respectively. Trypsin appeared as the clear zone on the blue background of the substrate gel.

#### 2.5.2. pH and Temperature Profile

The temperature profile of PT was studied at pH 8.0, using BAPNA as the substrate at various temperatures (25, 35, 45, 50, 55, 60, 65, and 70 °C). The pH profile of PT over a wide pH range (4.0–11.0) was investigated at 60 °C. The 50 mM acetic-acid–sodium-acetate buffer was used for pH 4–6; the 50 mM Tris-HCl buffer was used for pH 7.0–9.0, and the 50 mM glycine-NaOH buffer was used for pH 10.0–11.0. The optimum temperature for PT was used to determine the pH profile.

#### 2.5.3. pH and Thermal Stability

The pH and thermal stability of PT were studied under different pHs and temperature conditions. PT was incubated for 30 min under a wide pH range (4.0–11.0), with various buffers for different pHs, as mentioned above. The proteolytic reaction was conducted at 60 °C. For the thermal stability study, PT in 50 mM Tris-HCl, containing 10 mM CaCl_2_ (pH 8.5), was incubated at different temperatures (25, 35, 45, 50, 55, 60, 65, 70 °C) for 30 min. After 30 min of incubation in the specified conditions, the proteolytic activity was measured using BAPNA, as described previously.

#### 2.5.4. Effect of Inhibitors

The effects of various inhibitors on the activity of PT from Asian seabass were determined, according to the method of Khantaphant and Benjakul [33]. PT in 50 mM Tris-HCl buffer (pH 8.5) containing 10 mM CaCl_2_ was mixed with an equal volume of protease inhibitor solution, to obtain the final designated concentrations (0.1 mM E-64, 1 mM iodoacetic acid, 1.0 mg/mL SBTI, 5 mM TLCK, 5 mM TPCK, 0.01 mM pepstatin A, 1 mM PMSF, and 2 mM EDTA). The mixture was allowed to stand at room temperature (25–28 °C) for 15 min. The remaining activity was determined, and the percentage inhibition was then calculated. The control was conducted in the same manner, except that DW was used, instead of inhibitors.

### 2.6. Kinetic Studies

The kinetic studies of PT were carried out as described by Van Hau and Benjakul [38]. Substrates including BAPNA and TAME at various concentrations (0.01 to 2.0 mM) were used. The final enzyme concentration for the assay was fixed at 0.0050 mg protein/mL. The Michaelis–Menten constant (K_m_) and maximum velocity (V_max_) were calculated, using a Lineweaver–Burk double-reciprocal plot [39]. The following equation was used to determine the catalytic constant (*K*_cat_):*K*_cat_ = V_max_/[*E*]
where [*E*] denotes the molar concentration of the trypsin measured based on MW, calculated from the SDS-PAGE and protein concentration.

### 2.7. Determination of Protein Content

All samples were measured for their protein content, using the Biuret method [40]. Bovine serum albumin was used as the standard.

### 2.8. Extraction of Acid-Soluble Collagen (ASC) from Asian Seabass Skin

ASC was isolated from Asian seabass skin, following the method of Nilsuwan et al. [41]. The pretreated Asian seabass skin was swollen, using 50 volumes of 0.5 M acetic acid for 48 h 4 °C. After filtration, the filtrate was adjusted to pH 7.5, and salt at 2.6 M NaCl was added. After 1 h, a pellet containing ASC was collected via centrifugation (10,000× *g* for 5 min at 4 °C). The pellet was washed with DW for three cycles. The pellet was resuspended in DW, with a pellet/DW ratio of 1:50 (*w*/*v*), and homogenized at 5000 rpm for 30 s. The mixture was further stirred at 150 rpm for 30 min at 4 °C. After 30 min, the precipitate was centrifuged (10,000× *g*, 5 min, 4 °C), using a high-speed refrigerated centrifuge. After 3 cycles of the washing process, the pellets were collected and freeze dried (model CoolSafe 55 ScanLaf A/S, Lynge, Denmark). The ASC powder obtained was then further used in the preparation of hydrolyzed collagen.

### 2.9. Comparative Study on Hydrolysis of Acid-Soluble Collagen (ASC) by Partially Purified Seabass Trypsin (PPST) and Commercial Porcine Trypsin (CPT)

#### 2.9.1. Hydrolysis of ASC

ASC was hydrolyzed with PPST and CPT. Briefly, ASC (2 g) was mixed with 10 mL of distilled water, and stirred for 15 min at room temperature. The pH of the mixture was adjusted to 8.5, using 0.1 or 1 M NaOH. To initiate hydrolysis, PPST or CPT at a concentration of 5 unit/g of protein was added to the ASC mixture. The control was prepared similarly, without PPT or CPT addition. The reaction mixtures were incubated at 60 °C, on an orbital shaker installed in the water bath, for different times (0, 5, 10, 15, 30, 45, 60, 90, and 120 min). The hydrolytic reaction was terminated, via the addition of hot SDS (5%) to the reaction mixture at a ratio of 1:1 (*v*/*v*), and the mixture was kept in boiling water for 15 min. Thereafter, the reaction mixture was cooled down, and centrifuged at 8500× *g* for 15 min (Beckman Coulter, Allegra™ centrifuge, Temecula, CA, USA). The supernatants were collected and used for analyses.

#### 2.9.2. Degree of Hydrolysis (DH)

The method of Benjakul and Morrissey [42] was adopted to determine the DH of samples. The sample (125 μL) was mixed in 2 mL of 0.2 M phosphate buffer, pH 8.2, and 1.0 mL of TNBS (0.01%, *w*/*v*) solution. The reaction mixture was vortexed and incubated at 50 °C in the dark for 30 min. The reaction was then terminated with 2 mL of 0.1 M sodium sulfite. The absorbance was read at 420 nm. *L*-leucine (0.5 to 5.0 mM) was used as the standard for calculating the α-amino group content. The DH was computed as follows:DH = [(*L* − *L*_0_)/(*L*_max_ − *L*_0_)] × 100
where *L* is the amount of α-amino groups in the hydrolyzed sample. *L*_0_ is the amount of α-amino groups in the initial sample. *L*_max_ is the total α-amino groups obtained after acid hydrolysis (6 M HCl at 100 °C for 24 h).

#### 2.9.3. SDS–Polyacrylamide Gel Electrophoresis (SDS-PAGE)

The protein patterns of the hydrolyzed samples were examined using SDS-PAGE, as described above.

#### 2.9.4. Size Distribution of Hydrolyzed Collagen

The samples showing the highest enzymatic degradation were investigated using MALDI-TOF, as described by Benjakul et al. [43]. The Autoflex Speed MALDI-TOF (Bruker, GmbH, Bremen, Germany) mass spectrometer, equipped with a 337 nm nitrogen laser, was used.

### 2.10. Statistical Analysis

All experiments and analyses were performed in triplicate. All the acquired data were subjected to analysis of variance (ANOVA). Mean comparisons were conducted via Duncan’s multiple-range tests.

## 3. Results and Discussion

### 3.1. Purification of Trypsin from Asian Seabass Pyloric Caeca (ASPC)

The purification of trypsin from ASPC is summarized in Table 1. Ammonium sulphate (AS) precipitation at 40–60% saturation increased the purity by 8.6-fold, with an obtained yield of 33.3% (Table 1). However, further purification was necessary to remove the unwanted proteins and contaminated enzymes in the AS fraction. Therefore, the AS fraction was subsequently purified using the SBTI–Sepharose 4B affinity column (Figure 1). The elution profile showed a single peak at fraction No. 23, indicating the high specificity of the SBTI–Sepharose 4B affinity column to binding trypsin (Figure 1). After subjection to the affinity column, the purity of the trypsin increased by 18.5-fold, with a yield of 15.2%. The SBTI–Sepharose 4B affinity column has been documented as being highly specific to various trypsins, as evidenced by a single sharp elution peak, indicating the absence of unwanted proteins and other enzymes bound with the column [34,38,44]. Brownstripe red snapper trypsin was also purified using the SBTI–Sepharose 4B affinity column, with a 13-fold purity, and 16% yield [34]. Bigeye snapper trypsin showed a 34-fold purity, and a yield of 13%, whereas Unicorn leatherjacket trypsin showed a purity 26.43-fold, with a yield of 13.43% [44], when the SBTI–Sepharose 4B affinity column was employed for purification.

### 3.2. Sodium Dodecyl Sulphate–Polyacrylamide Gel Electrophoresis (SDS-PAGE) and Substrate-Gel Electrophoresis

The protein pattern of PPST showed several protein bands with different molecular weights (MWs) (Figure 2A, Lane 2). Most protein bands had a MW lower than 97 kDa. PT had a MW of 23.5 kDa, as determined by SDS-PAGE (Figure 2A, Lane 3). Native PAGE (Figure 2C) also showed a single protein band, reconfirming that the purified trypsin had only one isoform. In general, trypsins have MWs in the range of 20–42 kDa [27]. In particular, different MWs for trypsins have been documented in numerous fish species, including 21.7 kDa for trypsin from the *Cirrhinus mrigala* digestive tract [12], 23.2 kDa for trypsin from the *Clupeonella cultriventris caspia* pyloric caeca [13], 23.5 kDa for trypsin from the *Arapaima gigas* pyloric caeca [14], 24 kDa from trypsin from the *Scorpaena notata* intestine [15], 21 and 24 kDa for trypsin from the *Thunnus alalunga* liver [16], etc. Differences in the MWs of trypsin from various fish species might be due to varying factors, including genetic differences, the organs used, autolytic degradation, a different climate and habitat, etc. [27].

The protease activity of PT from the Sepharose 4B column fraction was demonstrated via substrate gel electrophoresis, as illustrated in Figure 2B. A clear band with a MW of 23.5 kDa appeared on the gel. The result confirmed the presence of protease (Figure 2B), with hydrolytic activity toward casein distributed in the gel. Thus, it was confirmed that a clear band with a MW of 23.5 kDa belonged to a trypsin with proteolytic activity.

### 3.3. pH and Temperature Profile

The effect of temperature and pH on PT from the pyloric caeca of Asian seabass was studied at different temperatures and pHs. PT showed optimum hydrolytic activity against BAPNA at 60 °C, but an increasing temperature reduced its hydrolytic activity (Figure 3A). Enzyme denaturation obviously occurred at temperatures higher than the optimal temperature of trypsin. This resulted in a decrease in hydrolytic activity [34]. Trypsins from various fishes have been documented to have an optimum hydrolytic activity at 55–60 °C. Albacore tuna spleen trypsin showed the maximum hydrolytic activity at 55 °C [45], and Unicorn leatherjacket pyloric caeca trypsin had an optimal temperature of 55 °C [44]. Trypsin from the pyloric caeca of the New Zealand hoki [46], brownstripe red snapper [34], and chinook salmon showed a maximum hydrolytic activity at 60 °C [47]. However, trypsins from cold-water fish showed an optimum hydrolytic activity at lower temperatures, ranging from 40 to 45 °C. Such differences in trypsin hydrolytic activity are most likely due to the different environmental and living conditions [48].

Trypsin is usually active at alkaline pH values (7.5 and 10.5) [49]. The optimum hydrolytic activity of PT from Asian seabass pyloric caeca was noticed at pH 8.5, while it was significantly decreased at pH 4.0 (Figure 3B). The trypsin was very active from pH 7.5 to 9.0. Nevertheless, a drastic decline in activity was recorded at over pH 9.0. Trypsin from other fish also showed high activity at alkaline pHs. Trypsin from bigeye snapper pyloric caeca [38] and Unicorn leatherjacket pyloric caeca [44] had an optimum activity at pH 8.0. Trypsin from spotted goatfish pyloric caeca [50] and albacore tuna spleen had an optimal pH at 9.0 [51]. This might be due to the denaturation of trypsin under a very alkaline pH, in which the repulsive force of the negative charge of the trypsin molecule was dominant. This plausibly caused the altered conformation, thus losing its activity.

### 3.4. pH and Thermal Stability

PT from the pyloric caeca of Asian seabass was highly stable at temperatures between 25 and 60 °C (Figure 3C). The PT had a residual activity level of 91.5% and 90.5% at 50 °C and 60 °C, respectively. The further increase in temperature from 65 °C to 70 °C caused a remarkable decrease in trypsin activity (Figure 3C). The intensive heat with the higher temperature most likely induced the denaturation of the PT, which negatively affected its hydrolytic activity. However, PT was very stable at higher temperatures than the trypsins from albacore tuna, Unicorn leatherjacket, bluefish and New Zealand hoki, which showed a drastic decrease in thermal stability at temperatures above 45 °C and 50 °C [44,46,47,51]. PT was stable up to 60 °C. Similarly, the trypsin from the brownstripe red snapper pyloric caeca was stable at 55 °C [34]. Moreover, the temperature stability of fish trypsin is also dependent on environmental conditions [52].

Fish trypsins are generally stable at alkaline pHs [34]. PT was stable at a broad range of pHs, including slightly acidic, neutral, and alkaline pHs (Figure 3D). The residual activity of PT was 88% at pH 4.0–6.0, while a residual activity of 91% was found at pH 7.0. The optimum pH stability was attained at 8.0. Moreover, PT showed the high residual activities of 93% and 88% at pH 9.0 and 10.0. Nevertheless, the residual activity dropped to 26% at pH 11.0. Similar to PT, several fish trypsins, including the New Zealand hoki [46], brownstripe red snapper [34], albacore tuna [51], bigeye snapper [38], bluefish [53], and Unicorn leatherjacket [44] have been reported to be stable at neutral and alkaline pHs with residual activities of 80–100%.

### 3.5. Effects of Various Protease Inhibitors on the Activity of Trypsin from the Pyloric Caeca of Asian Seabass

The effects of various protease inhibitors on the activity of PT are shown in Table 2. PT was almost completely inhibited (85.77–99.47% inhibition) by soybean trypsin inhibitor (SBTI, a serine protease inhibitor), phenylmethylsulfonyl fluoride (PMSF, a serine protease inhibitor) and TLCK (a trypsin-specific inhibitor). E-64 and iodoacetic acid (cysteine protease inhibitors), pepstatin A (an aspartic protease inhibitor), EDTA (a metalloprotease inhibitor), and TPCK (a chymotrypsin specific inhibitor) had very small inhibitory effects on trypsin activities (0.73–6.15%). Similar results for the inhibitory study of purified trypsin from the pyloric caeca of the brownstripe red snapper were reported by Khantaphant and Benjakul [34]. The result reconfirmed that the protease purified from the pyloric caeca of Asian seabass was most likely a trypsin.

### 3.6. Enzyme Kinetics

Enzyme kinetics, often known as the rate of enzyme activity, serves as a guide for the maximal application of a particular enzyme, and its concentration for a certain substrate [54]. The kinetic data of PT from the pyloric caeca of Asian seabass are tabulated in Table 3. The Michaelis–Menten constant (K_m_) of PT for hydrolyzing BAPNA was 0.0078, and it was 0.009 for TAME. The K_m_ represents the substrate concentration at which an enzyme achieves half of the V_max_. It is frequently linked to an enzyme’s affinity for a substrate (Engel, 1977). A higher K_m_ value denotes the lower affinity of the enzyme for the substrate [55]. As a result, PT from Asian seabass offered a better affinity, compared to trypsin from other fish species. PT had a lower K_m_ value for hydrolyzing BAPNA than those of trypsin from *P. disjunctivus* [56], *Sardinella aurita* [23], and *Cirruhinus mrigala* [12]. Conversely, PT had a higher K_m_ than trypsin from *Coryphaena hippurus* [20] when BAPNA was used as a substrate (Table 3). Similarly, PT had a lower K_m_ value for TAME hydrolysis than those of trypsin from *Macruronus novaezealandiae* [46], *Engraulis japonica* [57], and *Lutjanus vitta* [34], but showed a higher K_m_ than trypsin from *Thunnus tonggol* [58].

PT from the pyloric caeca of Asian seabass had a catalytic constant (k_cat_) of 5.4 for BAPNA, and 4.8 for TAME_._ K_cat_ is the maximum number of substrate molecules in which an enzyme can convert into a product per active site, when the substrate is saturated, in a given amount of time [55]. A higher K_cat_ demonstrates a greater conversion of the substrate into the product, indicating the better catalytic performance of the enzyme. PT showed a higher K_cat_ for BAPNA than the trypsin extracted from *P. disjunctivus* [56] and *Sardinella aurita* [23]. Nonetheless, PT had a K_cat_ lower than that of trypsin from *Cirruhinus mrigala* [12] and *Coryphaena hippurus* [20] (Table 3). PT had a relatively lower K_cat_ for TAME. The K_cat_ for TAME was 4.8, which was still higher than the K_cat_ of trypsin extracted from different fish species, including *Lutjanus vitta* [34], *M. novaezealandlae* [46], and *Thunnus tonggol* [58] (Table 3). PT had a relatively higher catalytic efficiency (K_cat_/K_m_) than trypsin from many fish species [12,23]. Catalytic efficiency shows how efficiently and specifically an enzyme binds to its substrate for certain reactions [59]. A higher catalytic efficiency means higher enzymatic activities for the conversion of the substrate into its corresponding product. The catalytic efficiency of PT for BAPNA was 69.22, and it was 53.46 for TAME. When BAPNA and TAME were used as substrates, PT showed a better catalytic ability than trypsin extracted from various fish species (Table 3).

### 3.7. Hydrolysis of Acid-Soluble Collagen (ASC) by Partially Purified Seabass Trypsin (PPST) and Commercial Porcine Trypsin (CPT)

#### 3.7.1. Degree of Hydrolysis (DH)

The DH of ASC hydrolyzed using PPST and CPT at 5 unit/g protein as a function of the time at 60 °C is displayed in Figure 4. The sample hydrolyzed using PPST and CPT showed different DHs as a function of the time. The DH of ASC was augmented with the increasing time of hydrolysis. Within the first 15 min, obvious hydrolysis was observed, followed by a lower rate of hydrolysis after 60 min. The results indicated that a greater number of peptide bonds were cleaved during the initial stage [60]. Similar results were observed when coconut milk was hydrolyzed by partially purified trypsin from Asian seabass [9]. The rate of hydrolysis was lowered after 30 min for CPT. Fewer cleavage sites, less enzyme auto-digestion, and/or less product inhibition were postulated [61]. The hydrolysis of ASC using PPST and CPT generated a number of free amino acids, and/or small peptides. At the same protease level and hydrolysis time, a higher DH was noticed in the ASC hydrolyzed using CPT than in the ASC hydrolyzed using PPST. The results revealed that CPT was more competent than PPST in the hydrolysis of ASC. Nevertheless, CPT has religious restrictions. Therefore, trypsin from the Asian seabass pyloric caeca could be employed for the hydrolysis of ASC, instead of CPT. However, a higher amount of PPST might be required to achieve the desired DH.

#### 3.7.2. Protein Pattern

The protein patterns of ASC hydrolyzed using PPST and CPT at a level of 5 units/g proteins, as a function of the time under non-reducing conditions, are shown in Figure 5. For the control (without protease), the distinctive protein bands of α1-chain (125 kDa), α2-chain (110 kDa), β-chain (230 kDa), and other high-molecular-weight components (HMC) were observed (Figure 5, Lane C). ASC from Asian seabass skin was type I collagen, which was composed of one α2-chain, and two identical α1-chains [62]. A drastic decrease in the α1-chain, α2-chain, β-chain, γ, and HMC was attained after 5 min of hydrolysis using PPSP and CPT, with the concomitant formation of low-MW peptides or proteins. The disappearance was more noticeable with the increasing hydrolysis time. However, the protein patterns for the ASC hydrolyzed using PPST and CPT at the same enzyme level were found to be different after hydrolysis at various times. The proteins were highly degraded in the ASC hydrolyzed using CPT, compared to using PPST. The obtained results were consistent with the DH (Figure 4). All the aforementioned proteins almost disappeared when ASC was hydrolyzed by CPT with an increasing hydrolysis time up to 30 min. All proteins disappeared after 60 min, as ASC was hydrolyzed with PPST. The results suggested that PPST had lower proteolytic activity than CPT at the same enzyme level and hydrolysis time. PPST and CPT derived from fish and porcine sources, respectively, plausibly had different cleavage sites that might govern different proteolytic abilities. Furthermore, peptides with different MWs were more likely generated. The results revealed that PPST and CPT effectively hydrolyzed ASC, and produced several low-MW peptides, which might possibly have certain biological activities. The peptides in hydrolyzed collagen could induce the proliferation of fibroblast cells associated with wound healing and skin nourishment [31]. Furthermore, those peptides could induce the formation of calcium nodules in osteoblast cells, related to bone strengthening [63]. Moreover, free amino acids or small peptides could be absorbed into the body easily.

#### 3.7.3. Size Distribution

The MWs of the peptides in collagen hydrolysates are shown in Figure 6. The spectra showed several peaks, indicating the presence of several peptides in the samples, with various MWs. The peptides generated after the enzymatic hydrolysis of collagen using PPST and CPT showed slight differences in terms of their distribution. This might suggest different cleavage sites for those two enzymes. The dominant peak in PPST and CPT hydrolysates represented peptides with MWs of 2992 Da and 2970 Da, respectively. PPST and CPT could generate peptides with different MWs. The results coincided with varying DHs and protein patterns, in which the cleavage of several peptide bonds was noticed, and various low MW peptides were detected after protein hydrolysis using PPST and CPT, at a level of 5 units/g protein (Figure 4 and Figure 5). Similar results were reported by Patil et al. [64], in which the hydrolysis of several proteins (whey protein isolate, kidney bean protein isolate, and threadfin bream fish mince) was conducted using skipjack tuna pepsin and commercial porcine pepsin generated peptides with different MWs. The results suggested that enzymes from the different organisms generated peptides with different MWs. The MW significantly affects the biological and functional properties of hydrolysates [65]. Many studies have demonstrated that low-MW peptides exhibited significant antioxidant, antitumor, and ACE-inhibitory activity [66,67,68,69].

## 4. Conclusions

Purified trypsin (PT) from the pyloric caeca of Asian seabass showed a molecular weight of 23.5 kDa. PT exhibited the highest hydrolytic activity toward BAPNA and TAME at 60 °C, and pH 8.5. PT was inhibited (99.47% inhibition) by soybean trypsin inhibitor, and the catalytic efficiency (Kcat/Km) of PT for BAPNA was 692.2; it was 534.6 for TAME. The partially purified seabass trypsin (PPST) could be used in the hydrolysis of ASC, but its hydrolysis efficacy was slightly lower than commercial porcine trypsin (CPT). However, PPST could hydrolyze collagen, and generate a number of small peptides or proteins, which might possess some biological activities. Therefore, Asian seabass trypsin could be a commercially important alternative to some proteases that have religious restrictions.

## Figures and Tables

**Figure 1 foods-12-02937-f001:**
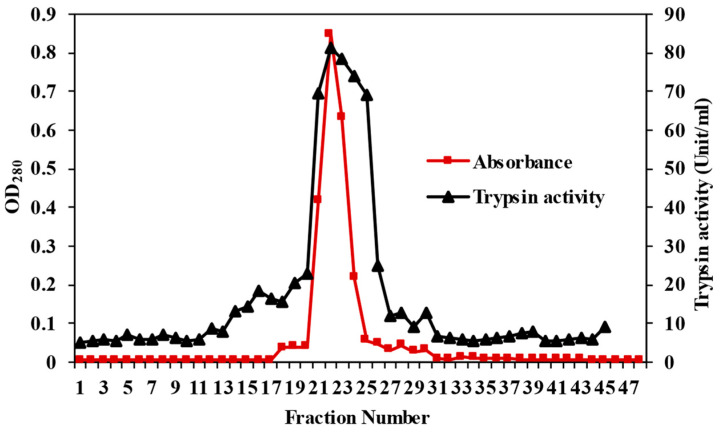
Elution profile of the ammonium sulphate (40–60% saturation) fraction on the SBTI–Sepharose 4B column. BAPNA was used as a substrate.

**Figure 2 foods-12-02937-f002:**
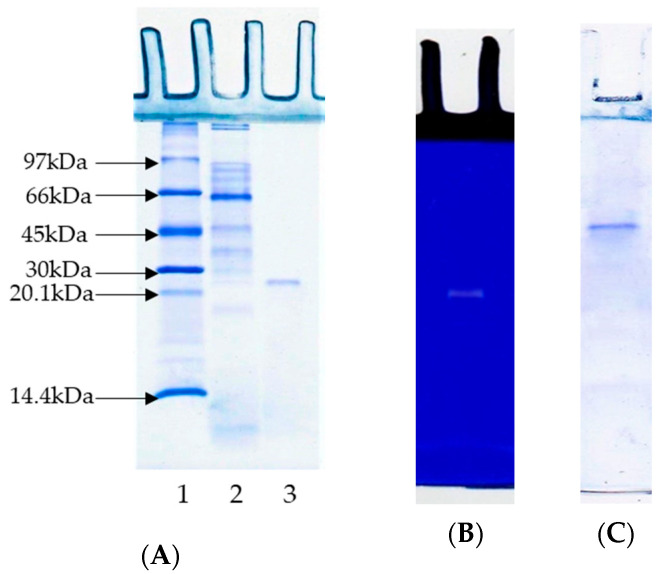
SDS-PAGE of trypsin from the pyloric caeca of Asian seabass (**A**). Lane 1: low-molecular-weight marker. Lane 2: ammonium sulfate fraction (40–60% saturation) of trypsin. Lane 3: purified trypsin from Sepharose 4B column, activity staining (**B**), and native PAGE of purified trypsin (**C**).

**Figure 3 foods-12-02937-f003:**
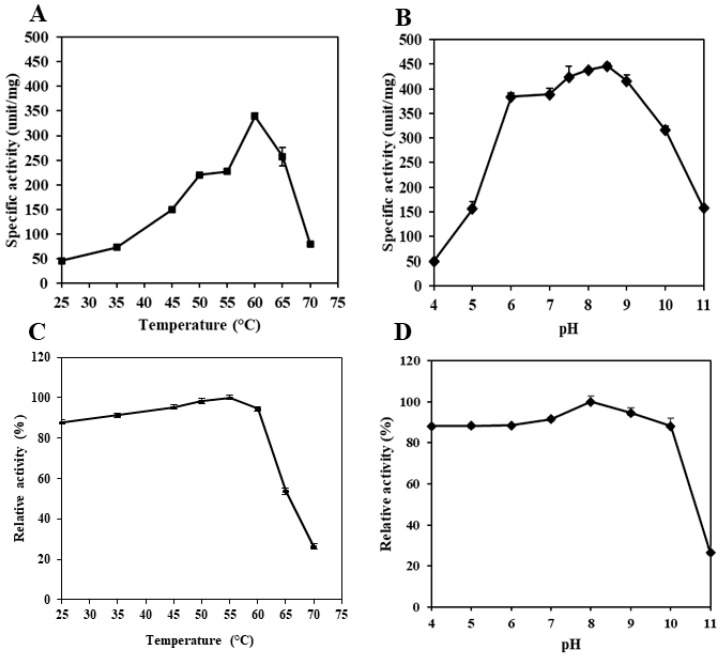
The temperature (**A**) and pH (**B**) profiles, and thermal (**C**) and pH (**D**) stability of purifiedtrypsin from the pyloric caeca of Asian seabass. For the temperature profile and stability, the activity of trypsin was determined at pH 8.0, using BAPNA as the substrate. The pH profile and stability were studied at 60 °C, and at various pHs.

**Figure 4 foods-12-02937-f004:**
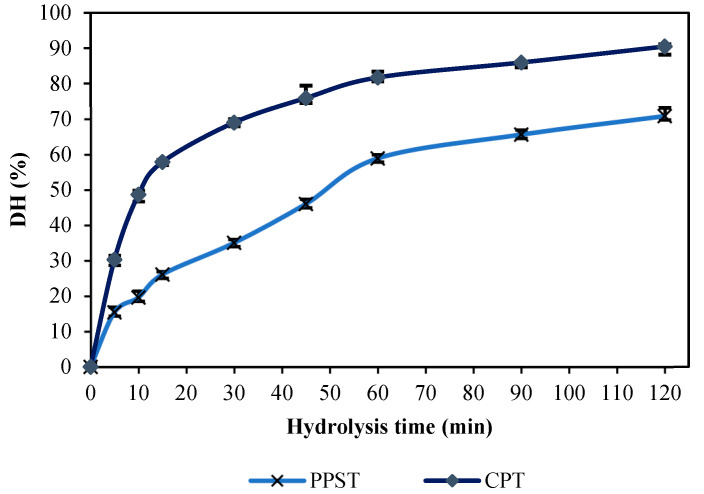
Degree of the hydrolysis (DH) of acid-soluble collagen (ASC) hydrolyzed using PPST and CPT, as a function of the hydrolysis time. Hydrolysis was conducted at 60 °C. PPST, partially purified seabass trypsin, CPT, commercial porcine trypsin. The level of trypsin used was 5 unit/g protein. Bars represent SD (n = 3).

**Figure 5 foods-12-02937-f005:**
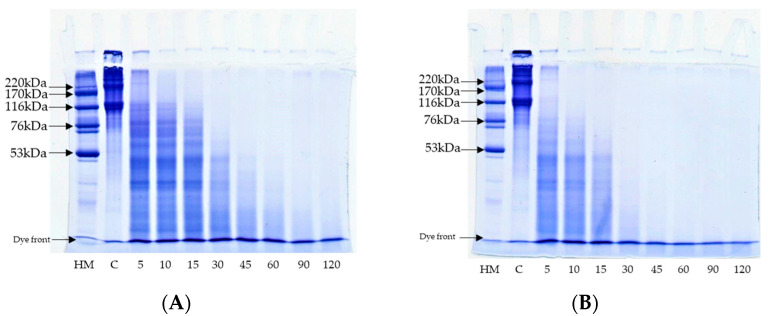
The protein pattern of acid-soluble collagen (ASC) hydrolyzed using PPST (**A**) and CPT (**B**) for different hydrolysis times. LM, low molecular weight; C, control without enzyme addition. Numbers designate the hydrolysis time (min). PPST, partially purified seabass trypsin; CPT, commercial porcine trypsin. The level of trypsin used was 5 unit/g protein.

**Figure 6 foods-12-02937-f006:**
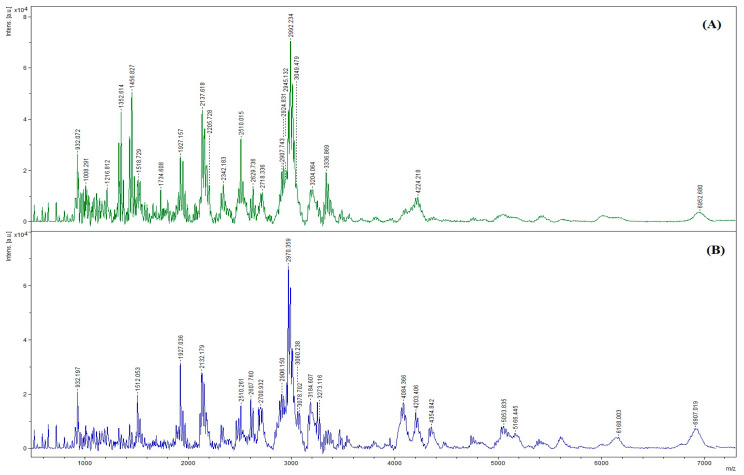
The size distribution of peptides in hydrolyzed collagen. (**A**) Acid-soluble collagen (ASC) was hydrolyzed by partially purified seabass trypsin (PPST). (**B**) ASC hydrolyzed using commercial porcine trypsin (CPT).

**Table 1 foods-12-02937-t001:** Purification of trypsin from the pyloric caeca of Asian seabass (*Lates calcarifer*).

Purification Fractions	Total Protein (mg)	Total Activity (unit ^a^)	Specific Activity (unit/mg)	Yield (%)	Purity (fold)
ASPC extract	296	5357	18.1	100	1
PPST	11.4	1783	156.4	33.3	8.6
PT	2.43	814	335	15.2	18.5

^a^One unit was defined as that releasing one nmol of *ρ*-nitroaniline per min, when BAPNA was used as substrate. ASPC, Asian seabass pyloric caeca; PPST, partially purified seabass trypsin; PT, purified trypsin.

**Table 2 foods-12-02937-t002:** Effects of various protease inhibitors on the inhibition of purified trypsin from the pyloric caeca of Asian seabass ^a^.

Inhibitors	Final Concentration	% Inhibition
E-64	0.1 mM	3.55 ± 0.89
Iodoacetic acid	1 mM	1.35 ± 1.01
SBTI	1 mg/mL	99.47 ± 0.17
TLCK	5 mM	93.62 ± 0.44
TPCK	5 mM	3.44 ± 1.75
EDTA	2 mM	6.15 ± 0.74
Pepstatin A	0.01 mM	0.73 ± 0.72
PMSF	1 mM	85.77 ± 0.92

^a^ The residual activities of trypsin after a 15 min incubation in the presence of various protease inhibitors were assayed at 60 °C and pH 8.5, using BAPNA as the substrate.

**Table 3 foods-12-02937-t003:** Kinetic parameters of trypsin from the pyloric caeca of Asian seabass (*Lates calcarifer*), and other fish trypsins.

Substrate	Sources of Trypsin	K_m_ ^A^ (mM)	K_cat_ ^A^ (s^−1^)	K_cat_/K_m_ (s^−1^mM^−1^)	References
BAPNA	*Lates calcarifer*	0.078	5.4	69.22	This study
	*Coryphaena hippurus*	0.035	241.04	6886.85	[20]
	*P. disjunctivus*	0.13	1.46	11.23	[56]
	*Sardinella aurita*	0.083	1.21	14.61	[23]
	*Cirruhinus mrigala*	0.0672	6.17	92.09	[12]
	*Lutjanus vitta*	0.507	4.17	9.27	[34]
TAME	*Lates calcarifer*	0.09	4.8	53.46	This study
	*Lutjanus vitta*	0.328	112	341	[34]
	*M. novaezealandlae*	2.08	19.00	9.1	[46]
	*Thunnus tonggol*	0.25	200	800	[58]
	*Engraulis japonica*	0.84	39.67	47.2	[57]

^A^ K_m_ and K_cat_ values of all trypsins were determined at 30 °C under optimal pH, except trypsins from *M. novaezealandlae*, which were assayed at 25 °C.

## Data Availability

The data presented in this study are available in the article.

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
