# Peer review of "Trypsin from Pyloric Caeca of Asian Seabass: Purification, Characterization, and Its Use in the Hydrolysis of Acid-Soluble Collagen"

_foods, 2023, doi:10.3390/foods12152937_

Round 1

Reviewer 1 Report

The manuscript demonstrate the purification of Trypsin from the pyloric caeca of Asian seabass through ammonium sulfate (40–60% saturation) precipitation and soybean trypsin inhibitor (SBTI)-Sepharose 4B column. A purification fold of 18.5 and a yield of 15.2% were achieved. SDS-PAGE analysis confirmed the presence of a single band of trypsin with a molecular weight of 23.5 kDa.

Explain likely reasons for the low recovery shown in the table.

It's suitable

Author Response

Responses to reviewer

Reviewer: 1

The manuscript demonstrate the purification of Trypsin from the pyloric caeca of Asian seabass through ammonium sulfate (40–60% saturation) precipitation and soybean trypsin inhibitor (SBTI)-Sepharose 4B column. A purification fold of 18.5 and a yield of 15.2% were achieved. SDS-PAGE analysis confirmed the presence of a single band of trypsin with a molecular weight of 23.5 kDa.

***** We appreciate your attention to the details of purification process for trypsin from the pyloric caeca of Asian seabass.

Explain likely reasons for the low recovery shown in the table.

***** Thank you for your insightful comment. Trypsin is a relatively unstable enzyme. There were 2 major steps of purification. Each process might cause the loss in activity of enzymes as indicated by the low yield as mentioned by the reviewer.

1) Ammonium sulfate precipitation: Ammonium sulphate (AS) precipitation at 40-60% saturation was used to recover the major trypsin. However, trypsin with other isoforms might not be recovered at this range of AS saturation. This was another reason for the low recovery or yield.

2) Affinity column: Soybean trypsin inhibitor (SBTI)-Sepharose 4B column was used.  After removing unbound trypsin, the elution of trypsin bound to the affinity column was performed using the acidic solution as the eluant. Although the receiving tube containing the alkaline solution for sudden neutralization was used for each fraction, some loss in activity could occur.  However, purified trypsin was obtained for further study, especially kinetic study, in which purified trypsin is required.

Reviewer 2 Report

This manuscript is devoted to an urgent problem related to the production of enzymes from the waste of the fishing industry. It is shown that the effectiveness of trypsin isolated from Asian sea bass is practically not inferior to the effectiveness of pork trypsin. It has been shown that the use of this enzyme is possible both in food and medical technologies. In addition, the use of products isolated from marine animals has no religious restrictions.

The authors propose a method for isolating trypsin from the pyloric appendages of Asian sea bass. For the obtained enzyme, some characteristics have been studied that allow it to be identified as an individual substance. The ability to hydrolyze collagen was studied for the obtained enzyme. Undoubtedly, these studies are relevant, as they simultaneously solve several modern problems: the search for new ways to isolate enzymes capable of hydrolyzing collagen; processing of waste from the fishing industry; obtaining products that do not have religious restrictions on their use in the food and medical industry.

The raw materials described in the manuscript were used by the authors for the first time. A detailed method of enzyme isolation is proposed in the section "materials and methods". Methods of analysis: electrophoresis, temperature dependences and calorimetry do not give reason to doubt the results obtained. Interestingly, fish collagen was chosen as the object of research of the proteolytic ability of trypsin, which was also obtained from fish production waste – the skin of Asian sea bass. Fish collagen-based products have a number of advantages over terrestrial animal collagen. It is 96% identical to human, and therefore does not cause allergic reactions. Fish are not carriers of diseases dangerous to humans. However, it is desirable to give a more detailed description of the release of collagen, a reference to the technique is not enough. The composition and especially the molecular weights of collagens isolated from fish skins may vary slightly. And the link [38] provides information for salmon. In the section "materials and methods" it is also a good idea to indicate the molecular weight characteristics of the isolated collagen.

The manuscript is a full-fledged study. The drawings are informative and do not require additions. The conclusions are justified. The list of references corresponds to the text of the article.

Author Response

Responses to reviewer

Reviewer: 2

This manuscript is devoted to an urgent problem related to the production of enzymes from the waste of the fishing industry. It is shown that the effectiveness of trypsin isolated from Asian sea bass is practically not inferior to the effectiveness of pork trypsin. It has been shown that the use of this enzyme is possible both in food and medical technologies. In addition, the use of products isolated from marine animals has no religious restrictions.

The authors propose a method for isolating trypsin from the pyloric appendages of Asian sea bass. For the obtained enzyme, some characteristics have been studied that allow it to be identified as an individual substance. The ability to hydrolyze collagen was studied for the obtained enzyme. Undoubtedly, these studies are relevant, as they simultaneously solve several modern problems: the search for new ways to isolate enzymes capable of hydrolyzing collagen; processing of waste from the fishing industry; obtaining products that do not have religious restrictions on their use in the food and medical industry.

***** Thank you for recognizing the novelty of our work. We sincerely appreciate your valuable time and careful evaluation of our manuscript. We are grateful for the positive remarks and constructive feedback provided. We are pleased to note that you realize the potential applications of trypsin in both food and medical technologies.

The raw materials described in the manuscript were used by the authors for the first time. A detailed method of enzyme isolation is proposed in the section "materials and methods". Methods of analysis: electrophoresis, temperature dependences and calorimetry do not give reason to doubt the results obtained. Interestingly, fish collagen was chosen as the object of research of the proteolytic ability of trypsin, which was also obtained from fish production waste – the skin of Asian sea bass. Fish collagen-based products have a number of advantages over terrestrial animal collagen. It is 96% identical to human, and therefore does not cause allergic reactions. Fish are not carriers of diseases dangerous to humans. However, it is desirable to give a more detailed description of the release of collagen, a reference to the technique is not enough. The composition and especially the molecular weights of collagens isolated from fish skins may vary slightly. And the link [38] provides information for salmon. In the section "materials and methods" it is also a good idea to indicate the molecular weight characteristics of the isolated collagen.

***** We deeply appreciate your thorough review and constructive feedback. Your expertise and detailed suggestion are absolutely invaluable in improving the manuscript. We are delighted that you found the detailed method of enzyme isolation satisfactory. Some experiments regarding characterization of trypsin have been added in the revised manuscript including Native-PAGE and effects of various protease inhibitors on the activity of trypsin from pyloric caeca of Asian seabass. Please see section 2.5.1., 2.5.4, 3.2 and 3.5. as well as line 277-279, 294.

The details regarding the extraction of acid soluble collagen from Asian seabass has been added in the revised manuscript. Please see section 2.8. The molecular weight characteristics of the isolated collagen has been added in section 3.6.2. Please see line number 425-426.

Moreover, size distribution data of peptides, generated after acid soluble collagen (ASC) hydrolyzed by partially purified seabass trypsin (PPST) and commercial porcine trypsin (CPT), has been added in the revised manuscripts to provide the detailed information on the peptides generated by two enzymes. Please see section 2.9.4 and 3.6.3.

The manuscript is a full-fledged study. The drawings are informative and do not require additions. The conclusions are justified. The list of references corresponds to the text of the article.

***** Thank you for your kind words and positive evaluation of our manuscript. We are happy to hear that you consider our work a full-fledged study and that the drawings were found to be informative and adequate. Once again, we extend our gratitude for your time and commitment to the peer-review process.

Reviewer 3 Report

In this paper, the authors thoroughly describe the extraction method of trypsin from the pyloric caeca of Asian seabass, and then analyze the characteristics of the extracted trypsin, including its protein molecular weight, pH and temperature effects, and kinetics. Finally, they compared the hydrolysis of acid soluble collagen using PPST (partially purified seabass trypsin) with commercial porcine trypsin. The results suggest that this particular trypsin has the potential to replace commercial proteases in the food industry. However, the following problems were identified:

1. In the introduction, it is mentioned that the molecular properties of trypsin from the pyloric caeca of Asian seabass have not been documented, but there is no research on molecular properties in the conclusion. It is also recommended to add a couple of sentences in the introduction to highlight the advantages of utilizing Asian seabass pyloric caeca for trypsin extraction. (Lines 68, 69)

2. In the Materials and Methods section, the experiment on pH stability should precede the pH profile to ensure that different buffer solutions used to adjust the pH of the reaction system do not affect enzyme activity. Furthermore, the study does not specify the pH during the investigation of thermal stability.

3. In 3.1. Purification of Trypsin from Asian Seabass Pyloric Caeca, the yield and purity of trypsin extracted from other marine organisms mentioned in the article should be supplemented in Table 1.

4. Figure 3 (B) indicates a remarkable decrease in enzyme activity within the pH range of 9-10. However, the text describes trypsin as highly active under alkaline conditions, from pH 8.0 to 10.0. (Lines 285, 286)

5. To streamline the content, consider reducing the number of references cited in Section 3.2 related to trypsin molecular weight and selecting the most representative ones.

6. In Table 2, the catalytic efficiency (Kcat/Km) of PT was mentioned as 69.2 for BAPNA and 53.4 for TAME. However, the article states values of 692.2 and 534.6, respectively. (Line 346)

7. The determination of protein content (2.7) and the statistical analysis (2.9) are not presented in the results.

8. It is recommended to adjust Figure 3 by removing the letters 'a b c d e' in the figure and modify Figure 2 and Figure 5 (A) to resemble Figure 5 (B) as well.

9. Please review the article for formatting and language problems and make the necessary corrections. (Lines 262, 285, 295)

Moderate editing of English language required

Author Response

Responses to reviewer

Reviewer: 3

In this paper, the authors thoroughly describe the extraction method of trypsin from the pyloric caeca of Asian seabass, and then analyze the characteristics of the extracted trypsin, including its protein molecular weight, pH and temperature effects, and kinetics. Finally, they compared the hydrolysis of acid soluble collagen using PPST (partially purified seabass trypsin) with commercial porcine trypsin. The results suggest that this particular trypsin has the potential to replace commercial proteases in the food industry. However, the following problems were identified:

*****Thank you for understanding our work. All queries have been responded and the corrections have been made in revised manuscript.

  1. In the introduction, it is mentioned that the molecular properties of trypsin from the pyloric caeca of Asian seabass have not been documented, but there is no research on molecular properties in the conclusion. It is also recommended to add a couple of sentences in the introduction to highlight the advantages of utilizing Asian seabass pyloric caeca for trypsin extraction. (Lines 68, 69)

***** Thank you for your comment. Molecular properties of trypsin from the pyloric caeca of Asian seabass including molecular weight, optimum pH and temperature of trypsin had been already included in the conclusion. Additionally, inhibitory studies and enzyme kinetics have been summarized in conclusion (revised manuscript). Please see line number 482-483.

Sentence regarding the advantages of utilizing Asian seabass pyloric caeca for trypsin extraction has been added in the introduction. Please see line number 73-76.

  1. In the Materials and Methods section, the experiment on pH stability should precede the pH profile to ensure that different buffer solutions used to adjust the pH of the reaction system do not affect enzyme activity. Furthermore, the study does not specify the pH during the investigation of thermal stability.

***** Thank you for your comment. For pH stability, the wide range of pH (4-11) with the aid varying buffers was used. Based on the pH profile, the activity showed the low activity at very acidic or alkaline pH. On the other hand, the pH stability, especially in acidic pH range, was quite stable. This was because the pH was readjusted to optimal pH before activity assay. The trypsin could be renatured as witnessed by the increased activity. However, trypsin is not commonly used or stored in acidic pH ranges, while the information obtained in the present study indicated that trypsin from Asian seabass should not be used at pH 11 or above.

The information on pH during the investigation of thermal stability has been added to the revised manuscript. Please see line number 174-175.

  1. In 3.1. Purification of Trypsin from Asian Seabass Pyloric Caeca, the yield and purity of trypsin extracted from other marine organisms mentioned in the article should be supplemented in Table 1.

***** Thank you for your suggestion. However, extraction methods and source of raw samples are different. Therefore, the yield and purity of trypsin extracted from some selected marine organisms had been mentioned in text instead of table. Thank you for your kind understanding.

  1. Figure 3 (B) indicates a remarkable decrease in enzyme activity within the pH range of 9-10. However, the text describes trypsin as highly active under alkaline conditions, from pH 8.0 to 10.0. (Lines 285, 286)

***** Thank you for your comment. Corrections have been made in revised manuscript. Please see line number 311-312. Discussion on activity loss at alkaline pH has been extended. Please see line 316-319.

  1. To streamline the content, consider reducing the number of references cited in Section 3.2 related to trypsin molecular weight and selecting the most representative ones.

***** 11 references have been removed in Section 3.2 and reference list as suggested by reviewer.

  1. In Table 2, the catalytic efficiency (Kcat/Km) of PT was mentioned as 69.2 for BAPNA and 53.4 for TAME. However, the article states values of 692.2 and 534.6, respectively. (Line 346)

*****Sorry for the typing error. The corrections have been made. Please see line number 391. Thank you.

  1. The determination of protein content (2.7) and the statistical analysis (2.9) are not presented in the results.

***** Thank you for your comment. Protein content had been already mentioned in the manuscript. Please see Table 1. Statistical analysis was not necessarily required in purification step. Following papers are few examples, where statistical analysis is not conducted for all the enzyme purification works:

https://doi.org/10.1016/j.foodchem.2006.05.050

https://doi.org/10.1021/jf060699d

https://doi.org/10.1016/j.foodchem.2008.11.087

https://doi.org/10.1016/j.foodchem.2009.09.098

  1. It is recommended to adjust Figure 3 by removing the letters 'a b c d e' in the figure and modify Figure 2 and Figure 5 (A) to resemble Figure 5 (B) as well.

***** Thank you for your suggestions. Corrections have been made as suggested by the reviewer. However, the changes in some figures might occur during submission process, which was out of our control. Sorry for this.

  1. Please review the article for formatting and language problems and make the necessary corrections. (Lines 262, 285, 295)

***** Thank you for your comment. Several corrections have been made for formatting and language problems throughout the manuscript. Also, authors have used ‘Grammarly’ software to check or English in the entire manuscript.

Reviewer 4 Report

We thank the authors of the paper for the good draft presented.

The aim of that work was to extract and characterize trypsin from pyloric caeca of Asian seabass and to compare its proteolytic ability toward acid soluble collagen with commercial porcine trypsin. Which can be a viable alternative to those on the market. The theme of the work is relevant. follow my few considerations:

I believe that the abstract does not represent the work in an excellent way, for example, not making clear the objective of finding an alternative way to produce an enzyme, especially trypsin, and the importance that this may have in the market. as well as exemplifying that this methodology has already been done in other species, I believe that the summary can be much better.

I believe that in the same way that I mentioned to value the summary, the induction could take this path.

Punctually:

L33: the term "religious" was not clear

L 37 "extracted from various marine sources", can be exemplified

Author Response

Responses to reviewer

Reviewer: 4

We thank the authors of the paper for the good draft presented.

We sincerely appreciate your positive feedback on our draft manuscript.

The aim of that work was to extract and characterize trypsin from pyloric caeca of Asian seabass and to compare its proteolytic ability toward acid soluble collagen with commercial porcine trypsin. Which can be a viable alternative to those on the market. The theme of the work is relevant. follow my few considerations:

We are pleased to hear that you find the theme of our work. Also, thank for your understanding in our research.

I believe that the abstract does not represent the work in an excellent way, for example, not making clear the objective of finding an alternative way to produce an enzyme, especially trypsin, and the importance that this may have in the market. as well as exemplifying that this methodology has already been done in other species, I believe that the summary can be much better.

*****Thank you for your suggestion. Necessary changes have been made for improvement of abstract to ensure it better represents the aim of our research. Specifically, highlighting the primary objective of finding an alternative enzyme and emphasizing its potential importance in the market. Please see line number 13-15 and 27.

I believe that in the same way that I mentioned to value the summary, the induction could take this path.

*****Thank you for your suggestion. An additional statement regarding the importance of seabass trypsin in the market has been included in the conclusion. Please see line number 482-483.

Punctually:

L33: the term "religious" was not clear

*****Thank you for your comment. ‘religious’ used in the text is the term related with ‘religions’. Generally, in Hinduism the cow is considered a sacred animal, and the consumption of beef is generally taboo. Whereas, pork is considered haram (forbidden) in Islam, and Muslims believe that consuming it is a sin. Therefore, both religions have limitations on food and food products derived from bovine and porcine. On the other hand, consumption of marine organisms and their products have no religious limitations.

Some change has been made. Please see line number 35.

L 37 "extracted from various marine sources", can be exemplified

*****Thank you for your suggestion. Examples of marine sources have been included in the revised manuscript with related references. Please see line number 38-41.
